# Out-of-School Sports Participation Is Positively Associated with Physical Literacy, but What about Physical Education? A Cross-Sectional Gender-Stratified Analysis during the COVID-19 Pandemic among High-School Adolescents

**DOI:** 10.3390/children9050753

**Published:** 2022-05-20

**Authors:** Mirela Sunda, Barbara Gilic, Damir Sekulic, Radenko Matic, Patrik Drid, Dan Iulian Alexe, Gheorghe Gabriel Cucui, Gabriel Stanica Lupu

**Affiliations:** 1Faculty of Kinesiology, University of Zagreb, 10000 Zagreb, Croatia; mirela.sunda@skole.hr (M.S.); bargil@kifst.hr (B.G.); 2Faculty of Kinesiology, University of Split, 21000 Split, Croatia; dado@kifst.hr; 3Faculty of Sport and Physical Education, University of Novi Sad, 21000 Novi Sad, Serbia; radenkomatic@uns.ac.rs (R.M.); patrikdrid@gmail.com (P.D.); 4Department of Physical and Occupational Therapy, Faculty of Movement, Sports and Health, Sciences, “Vasile Alecsandri” University of Bacau, 600001 Bacau, Romania; gabi.lupu@ub.ro; 5Department of Physical Education and Sports, Faculty of Humanities, “Valahia” University of Targoviste, 130070 Targoviste, Romania

**Keywords:** puberty, cognitive domain, affective domain, fitness status

## Abstract

Physical literacy (PL) is one of the main concepts related to lifelong physical activity (PA) and physical fitness (PF), but there is a lack of information on factors that might be associated with PL in adolescents from southeastern Europe. This study aimed to evaluate the associations between PF, participation and achievement in physical education (PE), out-of-school participation in sports, and PL in high school adolescents from Croatia. Participants were 298 high school students aged 14–18 years (191 females). Variables included school age, PE grade, sports participation, anthropometric indices, four PF tests, and PL (evidenced by CAPL-2-knowledge and understanding questionnaire (CAPL-2-KU) and PLAYself questionnaire). Gender-stratified analyses of differences were conducted using the t-test for independent samples or the Mann-Whitney test. Associations between variables were calculated with Pearson’s product moment correlation or Spearman’s rank order correlation. PF was positively correlated CAPL-2-KU in both genders. PE grade was significantly associated with PLAYself score (Pearson’s correlation = 0.36 and 0.38, *p* < 0.001 for boys and girls, respectively) but not with the CAPL-2-KU score. School age was not correlated with PL among boys, but there was a significant but negligible correlation between CAPL-2-KU and school age in girls (<2% of the common variance). Adolescents involved in sports had better PL and PF compared with adolescents not involved in sports. In conclusion, participation in out-of-school sports offers a good base for developing PL. Although this study took part over the COVID-19 pandemic period when the regular PE curriculum was significantly altered, the poor associations between school age and PL could lead to the assumption that the current PE curriculum does not allow for improvement of PL in later high school age, which warrants further investigation.

## 1. Introduction

Physical fitness (PF) is defined as a set of attributes that individuals have or achieve that is related to their ability to perform a physical activity (PA), and therefore, it is considered an integrated measure of most body functions, including musculoskeletal, cardiorespiratory, and metabolic systems [1]. Thus, PF is commonly observed as one of the key indicators of health status in children and adolescents [2,3]. At the same time, PA has a direct beneficial effect on various structures of PF; for instance, musculoskeletal and cardiovascular fitness and increased vigorous PA have regularly been found to be associated with higher cardiorespiratory and muscular fitness in children and adolescents [4,5]. However, increasing PA alone is considered insufficient for the continuous and lifelong maintenance of adequate PA and PF levels [6], and researchers suggest that there is a more complex concept in the background. In particular, the concept of physical literacy (PL) is theorized to be the gateway to achieving and maintaining sufficient levels of PA [7,8,9].

PL is a multidimensional concept that is defined as a “disposition to capitalize on our human embodied capability, wherein the individual has: the motivation, confidence, physical competence, knowledge and understanding to value and take responsibility for maintaining purposeful physical pursuits/activities throughout the life course” [10]. PL is a multidimensional concept that consists of four domains, including the physical, affective, cognitive, and social domains [11], and the majority of assessment tools are constructed to assess all four domains [12]. However, some assessments are designed to measure only certain domains of PL, with the domain of physical competence being investigated the most [12]. On the other hand, some researchers argue that the domain of physical competence of PL is over-emphasized and that the cognitive domain deserves increased attention [13]. Thus, researchers developed tools to assess certain domains of PL using questionnaires. PLAYself and the Canadian Assessment of Physical Literacy-2 knowledge and understanding (CAPL-2-KU) are some of the most popular instruments, intended to assess the affective (confidence and perceived competence) and cognitive (knowledge and understanding) domains of PL, respectively [14,15].

It is widely accepted that PL and PF should be intercorrelated, and studies have sporadically confirmed this relationship. In particular, a study on Canadian children aged 8–12 years noted strong associations between all domains of PL and cardiorespiratory fitness [16], and findings were similar among Spanish children aged 8–12 years [17]. Participating in sports activities is one of the best ways to improve PF, and it has been repeatedly proven that individuals involved in sports have better PF compared with nonactive individuals [18]. By this logic, it could be expected that adolescents involved in sports activities will possess higher PL levels than their nonactive peers [19]. This was proven indirectly in a study that reported that sports participation in childhood and adolescence was a predictor of participating in sports activities in adulthood [20]. However, studies directly investigating the interaction between sports participation and PL are generally lacking.

Apart from sports participation, physical education (PE) is another important influence on PL [21]. Indeed, PE in schools has the ideal position for fostering the development of PL, as its aim is to develop the skills, attitudes, and knowledge necessary for becoming a physically literate individual [13]. Thus, developing PL has become the main goal of PE in many countries, and it is embedded in PE curricula [13,22,23]. For instance, in evidence from a study on students aged 12–15 years from Ireland, interventions for improving PL during PE lessons led to an increase in several PL domains [24]. Similarly, a large study of Canadian teachers and students (8–15 years old) that focused on developing the knowledge of PE teachers to increase students’ skills, fitness, knowledge, and understanding related to PL showed increases in multiple domains of PL [25]. However, there is a clear lack of knowledge on possible associations between involvement in PE and PL in southeastern Europe, and to the best of our knowledge, no study has examined this issue in Croatia.

From the previous literature overview, it is clear that PL should be considered an important concept that is directly related to PF status. It is generally accepted that in adolescence, PL should be primarily developed through participation in sports and PE. However, while in some world regions, the levels of PL and factors that influence PL in children and adolescents are relatively well explored, there is a clear lack of studies that examined these issues in southeastern Europe. Therefore, this study aimed to evaluate the possible gender-specific associations between (i) PF, (ii) sports participation, and (iii) involvement/achievement in PE and PL among Croatian high schoolers. We hypothesized that (i) longer involvement and better achievement in PE and (ii) involvement in out-of-school sports would both be associated with better PL in high school adolescents from Croatia, irrespective of gender.

## 2. Materials and Methods

### 2.1. Participants

The research was conducted with 298 adolescents aged 14–18 years from Croatia (191 females, 16.19 ± 1.25 years; 107 males, 16.02 ± 1.23 years). All participants were attending high school and were of good health (i.e., they did not have any medical condition that would prevent them from participating in fitness tests). That is, only adolescents who were capable of performing fitness tests were included in the research, and adolescents who were ill or had any injury were excluded from the research. They were all participating in regular PE classes twice a week, while some of them were involved in competitive sports outside of the school curriculum. For the purpose of this study, participants were grouped in two groups by school year: younger, 1st or 2nd year of high school, and older, 3rd or 4th year of high school. Students had to sign an informed consent waiver before the study’s initiation, and parents or legal guardians had to sign a consent form for participants under 18 years of age. It is important to note that all participants were taught PE by the same teacher (the first author of the study) at the same high school in Dakovo, Croatia, which was intentional so as to avoid possible inconsistencies in the evaluation of the study variables (details are provided later in this paper) and to minimize the inevitable influence of differences in the school environment. The study was approved by the ethical board of the University of Zagreb, Faculty of Kinesiology (Ethical Board Approval No 25/2021).

### 2.2. Concept of the Research, Variables and Measurements

The concept of the research is presented in Figure 1.

In this study, we observed associations among five sets of variables: (i) school age (younger vs. older; please see previous section for details), (ii) students’ PE achievement (observed as PE grades at the end of the previous semester; five-point grade from 1 to 5), (iii) out-of-school sports participation (from school records: involved vs. noninvolved in out-of-school sports, (iv) PF status (using anthropometric and fitness indices), and (v) PL (measured as PLAYself and CAPL-2-KU scores) [15,26].

#### 2.2.1. Physical Fitness

Anthropometric variables included body height, body mass, and body mass index. Body height (BH) was measured in meters with an accuracy of 0.1 cm. During measurement, the students stood barefoot on a flat and firm surface in an upright position with their arms close to their bodies and looking straight ahead. For measuring body mass (BM), students stood barefoot on the digital diagnostic scale (Omron BF511), dressed in a sports T-shirt and shorts, and maintained an upright position. The result was read in kilograms within an accuracy of 0.1 kg. The body mass index (BMI) was calculated using the following formula: BMI = BM(kg)/BH(m)^2^ [27].

The fitness tests used in the study are part of the Croatian standardized fitness testing in schools: sit-and-reach test, sit-up test, standing broad jump, and a multilevel aerobic endurance test (beep test) [28]. The sit-and-reach test assesses the flexibility of the lower back and hamstrings muscles. The test is performed on a universal wooden bench. The students sit on a solid surface with their backs and buttocks leaning against a wall. Their legs are fully extended, and they touch the bench with the entire surface of their feet. The students extend their arms and perform the maximum forward bend. The result is the maximum reach of the hand, which is read in centimeters [29]. The sit-up test assesses the repetitive strength of the abdominal muscles. The students lie on their backs with their knees bent and feet placed on the floor. Their forearms and palms are placed on the upper leg. Once signaled to begin, the students perform the sit-up movement, with their palms gliding on the upper legs to the kneecaps and returning to the initial position. The result is recorded as the maximum number of properly executed sit-ups in 30 s [30]. The standing broad jump assesses the explosive power of the lower extremities. It is performed on a special mat (Ghia Sport, Pazin, Croatia) that measures jumps with a centimeter tape. The students stand behind the start mark with their feet in hip-width position with the top of the feet placed at the very edge of the starting line and perform a maximal forward jump. The length of the jump from zero on the centimeter tape to the footprint on the measuring pad closest to the point of reflection is measured. The result is read in centimeters [31]. The multilevel beep test is used to assess students’ aerobic endurance. It consists of running intervals over a 15 m court. With each level, the interval between sound signals decreases and the students must increase their running speed. The initial running speed is 8.5 km/h, and it increases by 0.5 at each level. An audio recording that emits sound signals is used to control the time intervals during the measurement and determine the running speed of the students. The result is expressed as the number of run levels and intervals. The test is performed in the school gymnasium on a flat surface [32].

#### 2.2.2. Physical Literacy

To estimate physical literacy, the CAPL-2-KU and PLAYself questionnaires were administered via the online platform SurveyMonkey (SurveyMonkey Inc., San Mateo, CA, USA). Two experienced researchers translated the original questionnaires into Croatian. Then, they were translated into English by a third researcher, and the native English speaker evaluated this final version. The Croatian versions of the questionnaires were finalized after correcting terms that were not understandable to two of the Croatian researchers.

The CAPL-2-KU questionnaire consisted of 12 questions related to knowledge and understanding of the importance of physical activity and its benefits and knowledge about cardiorespiratory endurance, fitness status, and muscle strength including guidelines for daily physical activity and sedentary time. The maximum number of points that can be achieved on the questionnaire is 12. Each question consisted of four answers, where the correct answer was rated with 1 and the incorrect answer was rated with 0. The feasibility, reliability, and validity of the questionnaire were confirmed in a study of Canadian children [15]. The Croatian version of the CAPL-2-KU questionnaire was used in this study.

The PLAYself questionnaire, a self-assessment questionnaire, was used to determine the adolescents’ self-perceived PL. The questionnaire consists of four groups of questions related to (i) environment (activities in different surroundings such as in the snow, water, or gym); (ii) a self-description of personal physical literacy (including questions such as, “I think I have enough skills to participate in all the sports and activities I want”, “I think that being active makes me happier”, and “I’m confident when doing physical activities”); (iii) the relative ranking of literacy, numeracy, and physical literacy; and (iv) fitness, which is defined by the item “My fitness is good enough to let me do all the activities I choose”; however, this question is not evaluated in the result. Subtotals from the first three groups divided by the number of questions represent the final result. The maximum score of 100 points indicates high self-perceived PL [33]. PLAYself demonstrated construct validity and good psychometric properties for children and youth [26]. The Croatian version of the PLAYself questionnaire was used in this study.

#### 2.2.3. Procedures

The measurements for this study were collected during two weeks in October 2021. Students were tested during two consecutive PE classes in sessions that lasted 90 min and were divided in groups of 20–30 according to their PE classes schedule. The testing was organized in the following way. During the first class, all students were measured on anthropometric indices. Afterward, they performed a warm-up and were tested on several fitness tests including the broad jump, the sit-and-reach, and sit-ups. The second testing day included PL assessment at the beginning of the class where students filled in the questionnaires on their mobile phones (or researchers provided phones for students who did not have a phone). Then, they performed a warm-up and were tested on the multilevel beep test. All measurements were conducted by two experienced researchers and two PE teachers who had rich experience in fitness testing and PL assessment.

### 2.3. Statistical Analysis

The Kolmogorov-Smirnov test was applied to identify the normality of the distributions for all variables; either frequencies and percentages or means and standard deviations were reported depending on the character of the variables.

To evaluate the reproducibility and reliability of the Croatian versions of the CAPL-2-KU and PLAYself, a subsample of 45 participants was tested in a test-retest procedure over a time frame of 7 days. The percentages of items with the same response, Cronbach’s alphas (CA), and inter-item-correlations (IIRs) were calculated.

Differences between age (younger vs. older) and involvement in out-of-school sports (involved vs. non-involved) were calculated with either a t-test for independent samples (for parametric variables) or the Mann-Whitney test (for nonparametric variables).

Correlations between variables were calculated with either Pearson’s product moment correlation (for parametric variables) or Spearman’s rank order correlation (for nonparametric variables).

Statistica ver. 13.5 (Tibco Inc., Palo Alto, CA, USA) was used for all analyses, and *p*-level of 0.05 was applied.

## 3. Results

The reliability of the CAPL-2-KU and PLAYself were appropriate to high, with Cronbach alphas ranging from 0.81 to 0.93, IIRs ranging from 0.76 and 0.82 (for test and retest, respectively), and percentages of items with the same number of responses of 80% and 89% (for the CAPL-2-KU and PLAYself, respectively).

Among the total sample not divided by gender, participants involved in out-of-school sports were taller and heavier and achieved better results for all PF variables except the sit-and-reach test of flexibility than their nonathletic peers. Further, athletic adolescents had higher CAPL-2-KU scores, higher scores on three of the five PLAYself subscales, and higher total PLAYself scores (Appendix A). Next, the participants in the older group were heavier, had higher BMI, and performed better on the broad jump and sit-up tests than those in the younger group, but there were no significant differences in PL between age groups (Appendix A). Participants’ age was positively correlated with body mass, BMI, three of four PF tests, and CAPL-2-KU score, while no significant correlation was found between age and PLAYself score. Additionally, participants with better PE grades were taller, had better PF, and achieved higher CAPL-2-KU and PLAYself total scores and PLAY environment and PLAY self-description subscale scores (Appendix A). Finally, for the total sample of participants, the PLAYself total score and the PLAY environment and PLAY self-description subscale scores were consistently positively correlated with the broad jump, sit-up, and beep tests with 10–25% of the shared variance (Appendix A).

When differences between groups based on sport participation were calculated only for boys, athletic boys outperformed their nonathletic peers on all PF tests except for the sit-and-reach. Boys involved in sport also achieved higher CAPL-2-KU and PLAYself total scores and higher PLAY environment and PLAY self-description subscale scores. Athletic girls were superior on all PF tests except the sit-and-reach and achieved higher PLAYself total scores as well as higher PLAY environment and PLAY self-description subscale scores than those of girls who were not involved in out-of-school sports (Table 1).

The boys from the older age group were taller and heavier, had higher BMI, and performed better on the broad jump and sit-up tests than did their younger peers. The older girls were heavier, had higher BMI, and achieved better results on the broad jump and sit-up tests than the younger girls (Table 2).

For boys, age was positively correlated with body height; mass; BMI; and broad jump, sit-up, and beep test results. In addition to the logical positive correlations between PE grade and PF indices (9–27% of the common variance), PE grade was positively correlated with CAPL-2-KU score; PLAYself total score; and PLAY environment, PLAY self-description, and PLAY physical literacy subscale scores (4–17% of the common variance). When calculating for girls only, age was positively correlated with body mass, BMI, sit-up achievement, and CAPL-2-KU. PE grade was negatively correlated with body mass and BMI (<5% of the common variance) and positively correlated with all fitness test results (4–18% of the common variance), the PLAYself total score, and the PLAY environment and PLAY self-description subscale scores (4–17% of the common variance) (Table 3).

Among boys, achievements on sit-ups and the beep test were positively correlated with PLAYself total score and PLAY environment, PLAY self-description, and PLAY physical literacy subscale scores (16–27% of the common variance). Among girls, PLAYself total score was significantly positively correlated to all PF tests (4–24% of the common variance), with additional significant positive correlations between PLAY environment and PLAY self-description subscale scores and PF indices (Table 4).

## 4. Discussion

With regard to the aims of this study, there are two primary findings. First, although there is a general increasing trend in PF with adolescents’ age, participants’ age (i.e., length of involvement in PE) was poorly associated with PL, with no significant differences between the younger and older adolescents in any of the PL variables. Second, adolescents involved in out-of-school sports had better PL and better PF compared with adolescents not involved in sports. Therefore, our initial study hypothesis is partially accepted.

### 4.1. Physical Literacy and Physical Education

The finding that older adolescents achieved better PF test results is logical and can be explained by the anthropometric, hormonal, and coordination changes with age (growth and development) [34]. However, despite the better fitness status in the older adolescents, we did not find evidence of significant differences in cognitive or affective domains of PL between age groups. Possible explanations are discussed in the following text.

The first explanation is based simply on the characteristics of the applied PL tests. In brief, PLAYself is a self-evaluated and perceived measure of PL [26]. Thus, it is possible that children were judging themselves in relation to their peers (i.e., their age group). Indeed, people tend to evaluate themselves according to their peers (gender, social class, involvement in the same sports), whereas children are usually exposed to peer influence because they are similar in many ways (i.e., age, gender, competence) [35]. According to social comparison theory, individuals evaluate themselves through comparison with others, and they commonly choose similar individuals with whom to compare themselves [36]. Additionally, children are good at relating themselves to and judging themselves against others and forming more precise and honest considerations of their abilities [37]. Therefore, it was expected that the observed adolescents in this study would not compare their PA with that of older/younger peers but only with that of their age-related peers, which could result in nonsignificant differences in the cognitive and affective domains of PL between age groups.

Despite the previous explanation, we must note that the generally poor correlation between age and PL (note that only in girls were age and CAPL-2-KU score significantly correlated and with less than 2% of the common variance) is in contrast with a study conducted of Canadian children aged 8–12 years that noted that higher PL was associated with increased age [15]. This disagreement between findings may simply be a result of differences in the participants’ ages in the two studies (i.e., we studied adolescents aged 14–18 years, whereas the Canadian study observed younger children aged 8–12 years). Therefore, it is possible that PL reaches its peak before the age of 14 years and does not change significantly later on.

However, another possible explanation for the lack of association between school age and PL also deserves attention. Briefly, it is logical to expect that PL will increase with age and education [16], but our results show that older students (who passed through more school years) did not achieve better PL results. Therefore, the lack of association between school-age and PL could be a consequence of inadequate education regarding PA and exercise guidelines in Croatian PE (e.g., lack of development of the cognitive domain of PL). Since the authors of this study are involved in a regular educational system, we can attest that the PL concept is not sufficiently provided throughout the PE curriculum, although it seems that this is not a problem solely in Croatia. Specifically, an English study reported that boys aged 14–16 years had limited understanding of PA guidelines, knowledge of PA’s importance, and knowledge of conducting muscular fitness activities, which are all important facets of overall PL [38]. Investigators explained that the school environment, with an emphasis on PE classes, does not provide adequate information on and guidelines for improving PA or theoretical knowledge of muscular fitness activities [38].

However, a lack of an association between PL and school age in PE could be at least partially a result of the fact that we observed students during the COVID-19 pandemic (please see Methods for details), when PE classes were merged into one 90-min class per week (instead of the regular two 45-min classes per week). This reorganization of the PE course undeniably and exponentially increased the demands on children. Namely, because of the dramatic decrease in PAL during the pandemic, teachers were often focused simply on PA itself [39]. Therefore, some regular teaching goals (e.g., increasing the awareness of PA in everyday life, improving theoretical knowledge on the effects of different types of physical exercise) were simply set aside. Now, PE teachers and creators of the curricula should be more focused on developing programs that will increase adolescents’ knowledge of performing PA every day and teach them the importance of PA.

### 4.2. Physical Literacy and Out-of-School Sports Participation

Sport is the main agent for developing PF, and adolescents who participate in sports activities regularly have better PF than their nonparticipating peers [40]. Therefore, our results of better PF in Croatian adolescents involved in sports were expected. However, for this study, it is more important to note that students involved in sports had better scores on both applied PL questionnaires than those of their peers who were not participating in organized sports during the study course.

The PLAYself is a measure of self-perceived competence and self-efficacy for engaging in PA, exercise, and sports. Although self-efficacy generally entails personal judgment about one’s abilities that form the basis for organizing and executing actions for achieving the desired performance [41], it was logical to assume that students involved in sports would have higher PLAYself scores. Supporting this, it has been reported that practicing sports was positively related to self-efficacy in 13- to 17-year-old students from Spain [42]. Moreover, Danish research noted that previous sports participation had positive associations with affective and physical domains of PL in older participants (18–34 years) [43]. Finally, a Canadian study noted that sports competence was related to PLAYself scores in children aged 8–14 years [26]. Thus, the fact that our participants who were involved in sports have higher perceived competence, confidence, and self-efficacy and overall higher self-evaluation of PL directly confirms the validity of the PLAYself questionnaire in Croatian high-school students.

The background of the (positive) association between sports participation and higher PL is relatively understandable. Namely, PL is developed through participation in various movement activities, including sports and exercise, and PL influences which activities are possible for adequate participation [10]. Thus, it is expected that active participation in sports and physical exercise will be associated with higher PL levels. Supporting this argument, a recent Finnish study concluded that individuals who were members of sports clubs engage in more general PA than nonparticipants, which highlights the fact that sports clubs have a substantial role in promoting knowledge of the importance of PA among adolescents [44]. However, the importance of participation in sports is not only highlighted in increased PA. For example, in a Finnish study, athletic adolescents showed higher health literacy [45], which is likely associated with the cognitive domain of PL (i.e., knowledge and understanding) [46]. Thus, it could be concluded that sports clubs promote health in that they most likely develop the knowledge of PA and exercise. Therefore, adolescents should be encouraged to participate in sports activities in not just a competitive but also a recreational manner. Indeed, the problem is that adolescents give up sports because they do not reach desired competitive results. The main goal of sports and health professionals should be promoting and increasing the percentages of adolescents who participate in sports for health and fun rather than for competition achievements.

### 4.3. Strengths and Limitations

As this study has a cross-sectional design, it must be noted that the relationship between sports participation and PL must be observed as bidirectional. Specifically, increased PL levels can be a consequence of practicing sports, but practicing sports can also lead to higher PL levels. However, looking at the results from both sides, it could be assumed that sport is a good medium for increasing PL and that increased PL leads to greater sports participation. Second, the sample of participants was observed in a specific period (i.e., the COVID-19 pandemic), and therefore, there is a possibility that alterations in the PE curriculum influenced the studied associations between PE achievement and PL as discussed previously. Third, this study explored the validity of the applied PL questionnaires in one specific sample (e.g., high school students). Therefore, results should be generalized to similar samples of participants.

The main strength of this study is that this is probably one of the first studies in Croatia and southeastern Europe that introduced the concept of PL and investigated cognitive and affective domains of PL in adolescents. Additionally, this study has great practical implications in terms of PE teachers being able to focus on developing educational materials for improving PL and consequently increasing overall PA among adolescents and positively influencing lifelong PA.

## 5. Conclusions

This study recorded that adolescents involved in sports had better PL and better fitness status compared with adolescents who were not involved in sports. Therefore, it seems that sports clubs offer a good base for developing habits of lifelong PA participation. In future studies, it will be important to determine the sports with greater and lesser influence on PL in adolescents.

On the other hand, school age (i.e., length of involvement in PE) was poorly related to cognitive and affective domains of PL in the studied sample. Moreover, PE grade was not associated with the cognitive (knowledge and understanding) domain of PL. Although this could be partially a result of the COVID-19 pandemic and alterations to the PE curriculum during this period, it is also possible that the PE curriculum does not provide adequate information that will consequently positively influence cognitive and affective PL among adolescents.

Therefore, our results suggest that a part of the PE curriculum in Croatia needs to be enriched and changed to involve students in PA and prepare them to use and apply skills and knowledge outside the PE classes and later throughout life. Specifically, it is necessary to enrich the PE curriculum aside from a simple execution of movement patterns. The PE curriculum could be broadened by incorporating elements that allow students to develop the knowledge and confidence to perform PA and exercise outside of class.

## Figures and Tables

**Figure 1 children-09-00753-f001:**
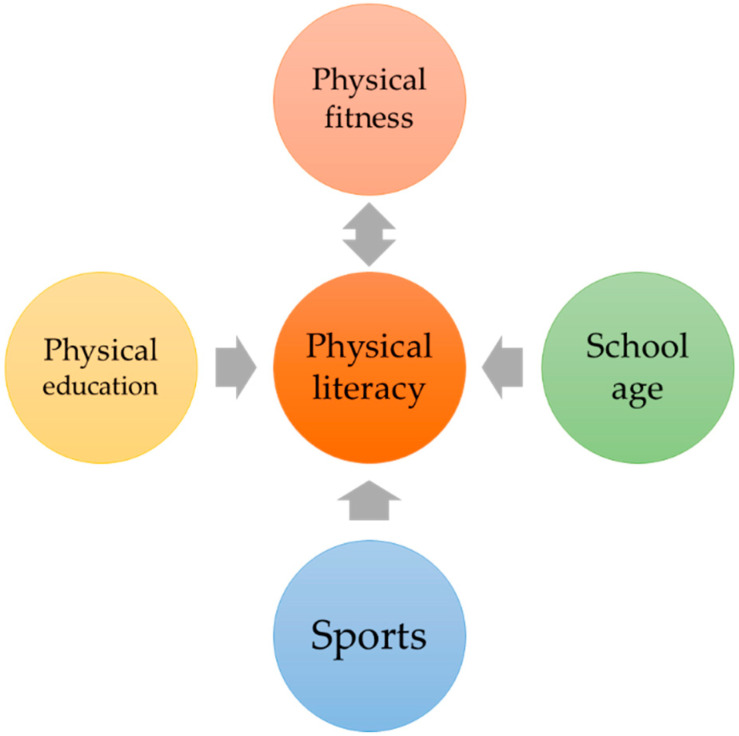
Concept and design of the study.

**Table 1 children-09-00753-t001:** Descriptive statistics and differences between groups based on sport participation (data are given as means ± SD).

	Boys		Girls	
	Athletic	Non-Athletic	Athletic	Non-Athletic
Body height (cm)	182.1 ± 6.48	181.13 ± 8.2	167.09 ± 6.53	168.4 ± 6.34
Body mass (kg)	75.58 ± 13.3	70.87 ± 15.01	58.84 ± 9.42	60.03 ± 10.78
Body mass index	22.75 ± 3.54	21.55 ± 4.15	21.06 ± 2.99	21.11 ± 3.16
Broad jump (cm)	216.81 ± 29.35	199.71 ± 28.8 ***	176.77 ± 22.66	162.73 ± 23.32 ***
Sit-and- reach (cm)	6.86 ± 8.18	5.47 ± 8.9	14.16 ± 6.21	13.63 ± 17.08
Sit-ups (repetitions)	68.46 ± 10.16	56 ± 9.46 ***	55.32 ± 10.53	50.11 ± 11.19 ***
Beep test (level)	12.18 ± 2.76	10.26 ± 2.45 ***	9.47 ± 2.59	7.67 ± 2.07 ***
CAPL-2-KU	9.13 ± 2.03	8.13 ± 2.72 *	9.25 ± 2	8.8 ± 2.02
PLAYself total	74.13 ± 9.76	63.06 ± 13.75 ***	74.89 ± 11.28	64.88 ± 12.31 ***
PLAY environment ^MW^	57.79 ± 16.12	45.61 ± 16 ***	56.88 ± 16.8	45.94 ± 16.87 ***
PLAY self-description ^MW^	82.06 ± 12.81	65.37 ± 16.71 ***	78.54 ± 15.52	64.12 ± 16.49 ***
PLAY literacy ^MW^	74.8 ± 19.72	70.22 ± 21.83	84.62 ± 16.19	83.86 ± 16.96
PLAY numeracy ^MW^	60.44 ± 25.14	65.55 ± 21.65	69.45 ± 22.39	64.52 ± 23.21
PLAY physical literacy ^MW^	88.16 ± 14.61	79.04 ± 18.22 *	92 ± 12.97	87.14 ± 17.35

Legend: ^MW^—differences between groups were calculated by Mann-Whitney test. *** *p* < 0.001, * *p* < 0.05.

**Table 2 children-09-00753-t002:** Descriptive statistics and differences between age groups (data are given as means ± SD).

	Boys		Girls	
	Younger	Older	Younger	Older
Body height (cm)	180.16 ± 7.31	183.51 ± 6.52 **	167.68 ± 6.71	168.13 ± 6.19
Body mass (kg)	71.39 ± 16.33	76.66 ± 10.48 *	57.65 ± 9.31	61.25 ± 10.85 *
Body mass index	21.91 ± 4.4	22.78 ± 2.96 *	20.48 ± 2.92	21.61 ± 3.15 **
Broad jump (cm)	200.13 ± 27.18	222.39 ± 29.18 ***	163.74 ± 21.6	171.19 ± 25.46 *
Sit-and- reach (cm)	5.93 ± 8.76	6.85 ± 8.1	12.09 ± 7.3	15.28 ± 17.93
Sit-ups (repetitions)	61.18 ± 11.48	67.18 ± 10.89 **	49.72 ± 10.26	53.9 ± 11.67 **
Beep test (level)	11.02 ± 2.94	12.02 ± 2.56	8.5 ± 2.41	8.16 ± 2.43
CAPL-2-KU	8.77 ± 2.39	8.78 ± 2.3	8.82 ± 2.06	9.08 ± 1.99
PLAYself total	69.15 ± 14.01	71.35 ± 10.54	69.11 ± 13	67.97 ± 12.77
PLAY environment ^MW^	52.31 ± 17.55	54.74 ± 16.54	50.14 ± 18.56	49.68 ± 16.84
PLAY self-description ^MW^	75.58 ± 18.57	76.73 ± 13.64	69.59 ± 16.85	69.11 ± 18.19
PLAY literacy ^MW^	70.3 ± 21.35	76.33 ± 19.27	83.78 ± 17.33	84.44 ± 16.12
PLAY numeracy ^MW^	61.17 ± 23.07	63.44 ± 25.12	67.46 ± 23.57	65.31 ± 22.53
PLAY physical literacy ^MW^	83.98 ± 18.57	85.95 ± 13.98	92.1 ± 14.14	86.17 ± 17.11 *

Legend: ^MW^—differences between groups were calculated by Mann Whitney test, *** *p* < 0.001, ** *p* < 0.01, * *p* < 0.05.

**Table 3 children-09-00753-t003:** Correlations between study variables and participants’ age and PE grade.

	Boys		Girls	
	Age	PE Grade	Age	PE Grade
Body height (cm)	0.19 *	0.09	0.03	0.03
Body mass (kg)	0.29 **	−0.10	0.23 ***	−0.16
Body mass index	0.20 *	−0.16	0.28 ***	−0.21 ***
Broad jump (cm)	0.47 ***	0.33 ***	0.13	0.44 ***
Sit-and- reach (cm)	0.15	0.16	0.13	0.18 **
Sit-ups (repetitions)	0.28 ***	0.56 ***	0.15*	0.39 ***
Beep test (level)	0.31 ***	0.39 ***	0.02	0.40 ***
CAPL-2-KU	0.09	0.19 *	0.21 ***	0.11
PLAYself total	0.09	0.36 ***	−0.05	0.38 ***
PLAY environment	0.08	0.28 ***	0.02	0.34 ***
PLAY self-description	0.01	0.42 ***	−0.08	0.42 ***
PLAY literacy	0.05	0.02	0.05	−0.05
PLAY numeracy	0.09	−0.04	−0.01	0.04
PLAY physical literacy	0.11	0.26 **	−0.12	0.05

Legend: *** *p* < 0.001, ** *p* < 0.01, * *p* < 0.05.

**Table 4 children-09-00753-t004:** Correlations between physical fitness tests and variables of physical literacy.

	Broad Jump		Sit-and-Reach		Sit-Ups		Beep Test	
	Boys	Girls	Boys	Girls	Boys	Girls	Boys	Girls
CAPL-2-KU	0.05	0.01	0.20	0.03	0.05	0.40 ***	0.02	0.02
PLAYself total	0.18	0.49 ***	0.03	0.17 *	0.51 ***	0.43 ***	0.47 ***	0.53 ***
PLAY environment	0.20	0.41 ***	0.06	0.15 *	0.42 ***	0.45 ***	0.41 ***	0.43 ***
PLAY self-description	0.20	0.55 ***	−0.02	0.18 **	0.54 ***	0.40 ***	0.53 ***	0.56 ***
PLAY literacy	−0.10	0.02	−0.02	0.03	0.12	0.43 ***	0.02	0.08
PLAY numeracy	0.00	0.09	0.00	0.06	−0.03	0.45 ***	−0.08	0.08
PLAY physical literacy	0.11	0.06	0.17	−0.01	0.36 ***	0.40 ***	0.35 ***	0.21 ***

Legend: *** *p* < 0.001, ** *p* < 0.01, * *p* < 0.05.

## Data Availability

Data is available here: https://www.dropbox.com/s/b6pqvv3sx0ye6pu/PL%20fitness%202%20matrica.sta?dl=0 (access on 15 March 2022).

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
