# Peer review of "Out-of-School Sports Participation Is Positively Associated with Physical Literacy, but What about Physical Education? A Cross-Sectional Gender-Stratified Analysis during the COVID-19 Pandemic among High-School Adolescents"

_children, 2022, doi:10.3390/children9050753_

Round 1

Reviewer 1 Report

In the introduction, find more relevant references that define the concept of physical literacy (PL), references 8 and 9.

In the methodology, should add references about the physical tests: sit-and-reach, sit-ups, the standing broad jump, the multilevel beep, as well as the CAPL-2-KU, and PLAYself questionnaires

In the results section, MW is shown in Tables 1, 2, and 3 for the variables with significant differences, but the variables from the physical tests are not included. In Tables 3 and 4, MW differences are mentioned and do not agree with the p < .05 value.

Author Response

In the introduction, find more relevant references that define the concept of physical literacy (PL), references 8 and 9.

RESPONSE: Thank you for this suggestion. We added more relevant references that define physical literacy:

  • Shearer, C.; Goss, H.R.; Edwards, L.C.; Keegan, R.J.; Knowles, Z.R.; Boddy, L.M.; Durden-Myers, E.J.; Foweather, L. How Is Physical Literacy Defined? A Contemporary Update. Journal of Teaching in Physical Education, 2019, 37, 237-245, doi:10.1123/jtpe.2018-0136.
  • Martins, J.; Onofre, M.; Mota, J.; Murphy, C.; Repond, R.-M.; Vost, H.; Cremosini, B.; Svrdlim, A.; Markovic, M.; Dudley, D. International approaches to the definition, philosophical tenets, and core elements of physical literacy: A scoping review. Prospects, 2021 50, 13-30, doi:10.1007/s11125-020-09466-1.
  • Shearer, C.; Goss, H.R.; Boddy, L.M.; Knowles, Z.R.; Durden-Myers, E.J.; Foweather, L. Assessments Related to the Physical, Affective and Cognitive Domains of Physical Literacy Amongst Children Aged 7–11.9 Years: A Systematic Review. Sports Medicine - Open, 2021 7, 37, doi:10.1186/s40798-021-00324-8.

Please see the 1st and 2nd paragraph in the Introduction.

In the methodology, should add references about the physical tests: sit-and-reach, sit-ups, the standing broad jump, the multilevel beep, as well as the CAPL-2-KU, and PLAYself questionnaires

RESPONSE: Thank you for this comment. We added references about the used tests. Text now reads: „...fitness tests including sit-and-reach tests, sit-ups, standing broad jumps, and multilevel beep tests; and a physical literacy assessment via PLAYself and CAPL-2-KU questionnaires [18,31]“;

„The body mass index (BMI) was calculated using the following formula: BMI = BM(kg)/BH(m)2 [32]“;

„The students extend their arms and perform the maximum forward bend. The result is the maximum reach of the hand, which is read in centimeters [34]“;

„The result is recorded as the maximum number of properly executed sit-ups in 30 seconds [35]“;

„The result is read in centimeters [36]“;

„The result is expressed as the number of run levels and intervals. The test is performed in the school gymnasium on a flat surface [37].“

Please see Materials and Methods, Concept of the research, variables and measurements section.

In the results section, MW is shown in Tables 1, 2, and 3 for the variables with significant differences, but the variables from the physical tests are not included. In Tables 3 and 4, MW differences are mentioned and do not agree with the p < .05 value.

RESPONSE: We checked the tables and corrected them. Also, we reduced the number of tables. Original version had 8 tables and current version has 4 tables, which sholud make the results section more clear.

Please see tables 1-4.

Staying at your disposal, thank you once again.

Reviewer 2 Report

The work presented addresses a topic of great interest and with social repercussion in general. But I recommend that authors address the following suggestions, before considering their publication.

I recommend that the authors better reflect the methodology in the abstract, they should remember that the abstract should include all the sections of the work.

The introduction addresses all those related to the bases to base the work, but they should reformulate it. I recommend that the authors substantiate it in a more general way. Paying more attention in the last paragraph that must be completely reformulated, I prioritize your hypothesis and finally the objective of the work in a clear and direct way.

In the methodology, I recommend that the authors write it in a chronological, orderly, schematized, and simple way.

Authors must specify what is in good health (line 120), as well as specify inclusion and exclusion criteria.

Reference in line 139 the validity of the questionnaires used.

They should better detail the characteristics of the instruments used, as well as the protocol applied. Detailing whether the data collection was done by a single person or several, if so, were they the same professional profile, and were they trained to avoid Inter explorer biases?

In the results section, I recommend that the authors present them in a more graphic and / or clear way.

The discussion longs for reflections from the authors on how these results would be used to improve adolescent physical activity.

The conclusions must clearly answer the objective that they reformulate again.

Author Response

The work presented addresses a topic of great interest and with social repercussion in general. But I recommend that authors address the following suggestions, before considering their publication.

RESPONSE: Thank you for recognizing the importance of our research and for providing your comments and suggestions. We tried to improve and ammend manuscript accordingly.

I recommend that the authors better reflect the methodology in the abstract, they should remember that the abstract should include all the sections of the work.

RESPONSE: Thank you for this comment. Text now reads: „Participants were 298 high school students aged 14-18 years (191 females). Variables included school-age, PE-grade, sports participation, anthropometric indices, four PF tests, and PL (evidenced by CAPL-2-knowledge and understanding questionnaire (CAPL-2-KU), and PLAYself questionnaire). Gender stratified analyses of differences were conducted using the t-test for independent samples or Mann-Whitney test. Associations between variables were calculated by Pearson’s product moment correlations or Spearman’s rank order correlation”.

The introduction addresses all those related to the bases to base the work, but they should reformulate it. I recommend that the authors substantiate it in a more general way. Paying more attention in the last paragraph that must be completely reformulated, I prioritize your hypothesis and finally the objective of the work in a clear and direct way.

RESPONSE: Thank you for this valuable suggestion. We changed the Introduction, made it 20% shorter and we believe it is now easier to follow. Introduction in the original version of the manuscript had 984 words, and now it has 804 words.

Also, we amended the last paragraph and hope that the objective of the riesearch is now more clear. Text now reads: „From the previous literature overview, it is clear that PL should be considered as important concept which in directly related to PF status. It is generally accepted that in adolescence the PL should be primary developed throughout participation in sports, and PE. However, while in some world regions the levels of PL, and factors that influence on PL in children and adolescents are relatively well explored, there is an evident lack of studies which examined these issues in southeastern Europe. Therefore, this study aimed to evaluate the gender-specific associations which may exist between (i) PF, (ii) sport-participation, and (iii) involvement/achievement in PE, and PL among Croatian high schoolers. We hypothesized that (i) longer involvement and better achievement in PE and (ii) involvement in out-of-school sports will both be associated with better PL in high school adolescents from Croatia, irrespective of gender“.

In the methodology, I recommend that the authors write it in a chronological, orderly, schematized, and simple way.

RESPONSE: Thank you for this comment, we tried to ammend it accordingly. We added a schematic view of the testing (Figure 1) and we organized variables description by separating it in several paragraphs (Paragraphs 2.2., 2.2.1, 2.2.2, 2.2.3). Also, we added a more detailed description about testing procedures. Text now reads: „The measurements of this study were collected during two weeks during October 2021. Students were tested during two consecutive PE classes that lasted 90 minutes and were divided in groups of 20-30 according to their PE classes schedule. The testing was organized in the following way. During the first class, all students were measured on anthropometric indices. After, they performed warm up and were tested on several fitness tests including the broad jump, sit-and-reach and sit-ups. The second testing day included PL assessment at the begging of the class where students filled in the questionnaires on their mobile phones (or researchers provided phones for students that did not have a phone). After, they performed warm up and were tested on the multilevel beep test. All measurements were conducted by two experienced researchers and two PE teachers, who had rich experience in fitness testing and PL assessment“. Please see Paragraph 2.2.3. Procedures.

Authors must specify what is in good health (line 120), as well as specify inclusion and exclusion criteria.

RESPONSE: We changed the text according to your suggestion. Text now reads: „All included participants were attending high school and were of good health (i.e., they did not have any medical condition that could prevent them from participating in fitness tests). Thus, only adolescents that were capable of performing fitness tests were included in the research, while adolescents that were ill or had any injury were excluded from the research“.

Please see Paragraph 2.1. Participants.

Reference in line 139 the validity of the questionnaires used.

RESPONSE: References have been added. Text now reads: „...and (v) PL (via PLAYself and CAPL-2-KU questionnaires) [15,26].“

Please see the 1st paragraph in the Concept of the research, variables and measurements section.

They should better detail the characteristics of the instruments used, as well as the protocol applied. Detailing whether the data collection was done by a single person or several, if so, were they the same professional profile, and were they trained to avoid Inter explorer biases?

RESPONSE: We explained about the fitness tests and PL questionnaires used and added relevant references, please see Variables and measurements section. Also, we added information about data collecting personel and testing procedures. Text now reads: „The measurements of this study were collected during two weeks during October 2021. Students were tested during two consecutive PE classes that lasted 90 minutes and were divided in groups of 20-30 according to their PE classes schedule. The testing was organized in the following way. During the first class, all students were measured on anthropometric indices. After, they performed warm up and were tested on several fitness tests including the broad jump, sit-and-reach and sit-ups. The second testing day included PL assessment at the begging of the class where students filled in the questionnaires on their mobile phones (or researchers provided phones for students that did not have a phone). After, they performed warm up and were tested on the multilevel beep test. All measurements were conducted by two experienced researchers and two PE teachers, who had rich experience in fitness testing and PL assessment “.

In the results section, I recommend that the authors present them in a more graphic and / or clear way.

RESPONSE: According to you suggestion we reduced the number of tables. Original version had 8 tables and current version has 4 tables, which sholud make the results section more clear. Also, we added a graph (please see Figure 1) that explains concept of the study which sholud also make it more clear.

The discussion longs for reflections from the authors on how these results would be used to improve adolescent physical activity.

RESPONSE: We added more clarification on guideliness for improving physical activity in adolescence. Text now reads:

„Thus, PE teachers and creators of the curricula should be more focused on developing programs that will increase the adolescents’ knowledge of performing PA everyday and teach them the importance of PA“.

„Therefore, adolescents should be encouraged to participate in sports activities not just in a competitive, but in a recreational manner. Indeed, the problem is that adolescents give up sports because they do not reach wanted competitive result. The main goal of sports and health professionals should be promoting and increasing the rate of adolescents that participate in sports for health and fun, and not for competition achieve-ments“.

Please see last paragraphs in the 4.1. and 4.2. sections.

The conclusions must clearly answer the objective that they reformulate again.

RESPONSE: The conclusion has been changed according to the objectives. Text now reads: „This study recorded that adolescents involved in sports had better PL and better fitness status compared to adolescents not involved in sports. Therefore, it seems that sports clubs offer a good base for developing habits of lifelong PA participation. In future studies it would be important to determine the type of sports with stronger and those with lower influence on PL in adolescents.

On the other hand, school age (i.e. length of the involvement in PE) was poorly related to cognitive and affective domains of PL in the studied sample. Moreover, PE grade was not associated with cognitive (knowledge and understanding) domain of PL. Although this could be partially a result of the COVID-19 pandemic and alterations to the PE curriculum in this period, it is also possible that the PE curriculum does not provide adequate information that will consequently positively influence cognitive and affective domains of PL among adolescents.

Therefore, our results suggest that a part of the PE curriculum in Croatia needs to be enriched and changed to involve students in PA and prepare them to use and apply skills and knowledge outside the PE classes and later throughout life. Specifically, it is necessary to enrich the PE curriculum; aside from a simple execution of movement patterns. The PE curriculum could be broaden by parts that allow students to develop the knowledge and confidence to perform PA and exercise outside of class. Please see Conclusion section.

Staying at your disposal, thank you once again.